# ID Insertion and Data Tracking with Frequency Offset for Physical Wireless Parameter Conversion Sensor Networks [note 1]

**DOI:** 10.3390/s19040767

**Published:** 2019-02-13

**Authors:** Osamu Takyu, Keiichiro Shirai, Mai Ohta, Takeo Fujii

**Affiliations:** 1Department Electrical & Computer Engineering, Shinshu University, Nagano 380-8553, Japan; shirai@cs.shinshu-u.ac.jp; 2Department Electronics Engineering and Computer Science, Fukuoka University, Fukuoka 814-0180, Japan; maiohta@fukuoka-u.ac.jp; 3Advanced Wireless and Communication Research Center (AWCC), The University of Electro-Communications, Tokyo 182-8585, Japan; fujii@awcc.uec.ac.jp

**Keywords:** frequency offset, interference cancellation, wireless sensor networks, data tracking

## Abstract

As the applications of the internet of things are becoming widely diversified, wireless sensor networks require real-time data reception, accommodation of access from several sensors, and low power consumption. In physical wireless parameter conversion sensor networks (PhyC-SN), all the sensors use frequency shift keying as the modulation scheme and then access the channel to the fusion center, simultaneously. As a result, the fusion center can recognize the statistical tendency of all the sensing results at a time from the frequency spectrum of the received signal. However, the information source, i.e., the sensor, cannot be specified from the received signal because no ID-indicating sensor is inserted to the signal. The data-tracking technique for tracing the time continuity of the sensing results is available for decomposing the sequence of the sensing results per sensor but the error tracking, which is a wrong recognition between the sensing results and the sensor, occurs owing to the similarity of the sensing results. This paper proposes the sensing result separation technique using a fractional carrier frequency offset (CFO) for PhyC-SN. In the proposed scheme, the particular fractional CFO is assigned to each user and it is useful for the ID specifying sensor. The fractional CFO causes inter-carrier interference (ICI). The ICI cancellation of the narrowband wireless communications is proposed. The two types of data-tracking techniques are proposed and are selectively used by the fusion center. Since the proposed data-tracking technique is multi-dimensional, high accuracy of data tracking is achieved even under the similar tendency of the sensing results. Based on computer simulation, we elucidate the advantage of the proposed sensing results separation.

## 1. Introduction

The Internet of things (IoT) is gaining considerable attention because it can be applied to various situations of life and industry [1]. Wireless sensor networks (WSN) construct the infrastructure of transferring a state condition, which is the sensing result, from each sensor to the fusion center (FC). Therefore, as the applications of IoT are diversified, the requirements of WSN are also diversified.

Examples of the requirements include a long life for long driving of sensors, an accommodation of accepting the wireless access from several sensors [2], and real-time recognition of all the sensing results at a time [3]. A packet access selected by WSN could not satisfy these diversified requirements. In the packet access, an ALOHA and a carrier sense multiple access (CSMA) are used as distributed wireless packet access schemes. To avoid packet collision, the random time space, which is random back off, is set before starting the packet access but it causes delay. In addition, as the number of sensor nodes increases, the coincident collision occurs more frequently and thus the repeat transmission of the data packet causes additional delay [4]. The data size of the sensing results is equal to or smaller than the control information included in the header of the packet. Therefore, a signaling overhead of the control signal is not negligible and thus the power consumption of the sensor node and the usage efficiency of the frequency spectrum are degraded. For supporting the real-time data reception, the accommodation of several sensors, and the long life of sensors, a novel access protocol is required.

The authors propose physical wireless parameter conversion sensor networks (PhyC-SN) for achieving the real-time sensing of data collection [5]. One type of PhyC-SN is a frequency shift keying as a modulation scheme of sensing results. All sensors send the frequency modulated signal to the FC and then the FC detects the frequency spectrum of the received signal using a discrete Fourier transform (DFT).

In the frequency spectrum, the DFT’s number of the frequency spectrum with large power indicates the frequency shift value selected by the sensor. FC can recognize all the sensing results from the frequency spectrum at a time. Therefore, it can evaluate the median and the outlier of all the sensing results. Since PhyC-SN does not require the wireless access control, the highly real-time access is achieved. However, two problems regarding PhyC-SN exist. The first problem is to be unable to specify the sensor node from the sensing result. Since PhyC-SN does not send any ID for the specifying sensor, the signaling overhead is reduced but the FC cannot specify the sensor that sends each sensing result to it. The second problem is a CFO caused by the frequency mismatch between the transmitter and the receiver. In DFT analysis, the orthogonality of the subcarrier components is distorted and thus an ICI occurs. ICI makes the frequency spectrum sent by the other sensor masked and thus the FC fails to detect the sensing result. In addition, the false alarm indicating FC wrongly recognizes the ICI or the noise component as the sensing data sent by sensor occurs [6].

For separating the sensing results into the sequence of sensing result with common sensor node, a data-tracking technique applied to PhyC-SN is considered. The time tracking technique based on Kalman filter is considered for the separation of the sensing results [6]. After the sensing results are received for a certain duration, a maximum likelihood sequence detection separates the sequence of the sensing result in terms of time continuity [7]. These techniques include the following tasks. If certain sensing results are near, the separated sequence wrongly includes the sensing results with the different sensor node. This failure of separation is an error tracking. Although the sequence of the sensing results with common sensor node is successfully separated, the FC does not specify the sensor node from the separated sequence, but each sensor requires to periodically inform the FC of the relationship between the sensing result and the ID of the sensor node.

For recovering these tasks of data separation in PhyC-SN, this paper proposes the novel data separation and the informing scheme of the sensor node’s ID. In the proposed scheme, the particular CFO whose frequency shift size is smaller than the minimum frequency resolution of DFT is assigned to each sensor, where it is referred to as fractional CFO. Each sensor shifts the carrier frequency of the local oscillator by the assigned fractional CFO. As a result, the FC can recognize the fractional CFO as the sensor node’s ID from the received signal. It can estimate the fractional CFO by evaluating the phase rotating speed between the two or more signals detected via DFT. For avoiding ICI caused by fractional CFO, this paper proposes an ICI cancellation. For constructing an ICI replica, the channel state information (CSI) of wide frequency spread ICI components should be estimated. This paper proposes the construction of an ICI replica composed of the CSI estimated from the carrier frequency band.

As Ref. [6] uses data tracking for the separation of sensing results, this paper considers the data tracking based on not only sensing results but also CSI and CFO. If the sensor node and the wireless environment are static, the CSI demonstrates the time continuity [8]. In addition, a phase lock loop of the local oscillator also tracks certain carrier frequency [9] and the CFO can be maintained by certain time duration. The proposed cost function for data tracking is composed of three dimensions, sensing result, CFO, and CSI. FC can determine the connection of the two-time continuous sensing results via the basement of multiple time continuities. As a result, the accuracy of data tracking is improved.

If multiple sensors select the common frequency number, which means these take the common sensing results, the CSI and CFO of each sensor cannot be estimated, separately. This is because the multiple access interference (MAI) is so significant that the estimation accuracy of the CSI and CFO is degraded. It is referred to as duplication of sensing results that multiple sensors take the common sensing results. For recovering the duplication of the sensing results, this paper proposes a duplication detector and the tracking based on the frequency spectrum. FC can count the types of received frequency spectrum with large signal power using ICI canceller. If the counted types of frequency spectrum are fewer than the number of sensors, FC can recognize that the duplication of sensing results occurs. The frequency spectrum in the duplication of the sensing results and is composed of the linear combination among the signals of the sensor nodes considering the common sensing results. If the sensor nodes considering the common sensing results are determined via FC in the last data transmission, the frequency spectrum of the sensing results can be recovered. As a result, the similarity between the received spectrum and the recovered one can be evaluated. It can be applicable to data tracking. The phase and amplitude of the frequency spectrum are determined using the CSIs and CFOs of all the sensors considering the common sensing results. This paper proposes the cost function based on the sensing result and the frequency spectrum for data tracking.

The progress of this paper compared with conventional papers is as follows.This paper proposes the fractional CFO used as the ID for specifying the sensor node from the received signal.For the ICI caused via CFO, the construction of the ICI replica from the narrow band wireless communication is proposed.The multiple dimensional tracking based on sensing result, CSI, and CFO is proposed. In addition, if the duplication of sensing results occurs, instead of it, the tracking based on sensing results and the spectrum is proposed.

Based on computer simulation, we explain the accuracy of the proposed data separation.

## 2. Related Works

The countermeasure to the problem of CFO in the wireless access from multiple sensors to the FC is considered to be that in the uplink cellular system.

In [10], the estimation of the CFO in uplink LTE is proposed by evaluating the phase rotating speed of the spectrum detected by a fast Fourier transform (FFT). For suppressing MAI, the estimated CFO is averaged using multiple subcarriers. Since the dynamic range of the received signal power among the users is significantly high, the suppression of MAI by averaging the estimated CFO is not effective. The estimation scheme of CFO with the suppression of MAI via interference cancellation is proposed [11]. A signal subspace analysis based on MIMO suppresses the MAI and then it estimates CFO [12,13] assuming that the estimation of CSI is ideal. The degradation of estimating CSI causes significant degradation of accuracy of subspace analysis. Since the impact of CFO is modeled by the cyclical convolutional operation, after compensating the impact of CFO, the residual MAI is suppressed by interference cancellation [14]. The extension version of it is constructed using the estimation of CSI [15]. These techniques require multiple antenna reception and the iterative processing and thus the processing delay and the complexity of the process are considerably large. The estimation technique of both CFO and CSI based on maximal likelihood sequence detection (MLSE) is considered [16] but its computational complexity is large. A self-cancellation technique for suppressing the MAI caused by CFO is proposed [17,18] but the usage efficiency of the frequency spectrum is reduced by half.

Various compensation techniques of CFO for WSN have been considered under the assumption of narrow band wireless communication systems. Recently, a low power wide area (LPWA) is gaining considerable attention as the long-range wireless sensor network. In [19,20], the random-access channel (RACH) for narrow band IoT uses the frequency hopping scheme as the wireless access protocol and thus the suppression of MAI and the estimation of CFO through RACH are considered. In [21], the frequency synchronization scheme based on global positioning system for LPWA is considered. The unique word for estimating CFO is considered [22]. In [23], the lower bound of the accuracy of CFO estimation in the single carrier communication is derived. In the compensation of COR for LPWA, the pilot signal for estimating CFO is required. Since the data rate of LPWA is small, the overhead of inserting pilot signal to the payload is large.

From these conventional considerations, the suppression of MAI is an important task under the multiple access environment, which is in the access from multiple sensors to a FC. The interference cancellation is one of the most powerful techniques for suppressing MAI. Since MAI is widely spread owing to CFO, the estimation of CSI in the wide band channels is necessary. However, WSN, such as LPWA and PhyC-SN, is narrow band communication and thus the overhead of the pilot signal for wide band channels is much large. In addition, the construction of MAI and the ICI replica is necessary, but any other paper does not consider it in the narrow band communications.

This paper proposes the estimation of CSI and CFO using data signal in PhyC-SN. The overhead for estimating CSI and CFO is limited. This paper also considers the construction of the wide spread ICI from the signal of the narrow band communication. This construction has not been considered yet in the WSN including LPWA. In addition, the insertion of ID to the data channel is proposed using fractional CFO in PhyC-SN. The overhead of inserting ID is limited. The multiple dimensional tracking for the highly accurate data separation is proposed. As a result, the recognition of individual data can be achieved via the PhyC-SN with the ID insertion of small overhead.

## 3. Overview of PhyC-SN

### 3.1. Data Transmission and Collection in PhyC-SN

Figure 1 shows the system model of the considered WSN. Since multiple sensors access to a FC via wireless communication, a star type network topology is assumed. Sensing results are modeled as multi-level discrete information. The transmitter of PhyC-SN transmits the continuous carrier whose frequency is a discrete value decided by inverse discrete Fourier transform (IDFT) and is selected in accordance with the conversion table between the discrete frequency and the sensing result, where the carrier with discrete frequency is referred to as subcarrier similar to an orthogonal frequency division multiplexing (OFDM). For example, if the sensing results are at fifth level, the sensor sends the fifth subcarrier to the FC. The modulation scheme of PhyC-SN is the same as a frequency shift keying. The maximal levels of sensing results are equal to the number of subcarriers. All the sensors send subcarrier to FC, simultaneously. Therefore, FC receives the signal composed of the mixed subcarriers.

When the *k*th sensor (k=1,2,…,K and *K* is the total number of sensors) sends $n_{k}$th subcarrier ($n_{k}\in 1,2,\ldots ,N$ and *N* is the total number of subcarriers) to the FC, the transmit signal in the *m*th time interval is decided using IDFT as follows.(1)$$\begin{array}{c} x_{k}\left(m\right)=\exp\left(j2\pi \frac{n_{k}m}{N}\right) \end{array}$$

Wireless communication is modeled as the multipath fading with *L* paths and independent identical distribution for each path and each user. The impulse response and the delay time in the *l*th path of the *k*th sensor are $h_{k,l}$ and $\tau_{k,l}$, respectively, where $\tau_{k,l}$ includes the time difference caused by detection timing mismatch.

In *m*th time interval, the signal received by the FC is given as follows.(2)$$\begin{array}{c} y\left(m\right)=\sum_{k=1}^{K}\sum_{l=1}^{L}h_{k,l}\exp\left(j2\pi \frac{n_{k}m-n_{k}\tau_{k,l}+m\varepsilon_{k}}{N}\right)\\ +n_{0}\left(m\right), \end{array}$$where $n_{0}\left(m\right)$ is a noise component and $\varepsilon_{k}\;\left(-1.0<\varepsilon <1.0\right)$ is the CFO derived by the frequency mismatch of local oscillator between *k*th sensor and FC. The value of CFO is normalized by the frequency interval of subcarriers.

FC analyzes a frequency spectrum of received signal by DFT. In PhyC-SN, all the sensors send the continuous wave for certain time duration. We assume the access timing difference among the sensors to be smaller than a time duration of DFT detection. Since DFT can detect symbol duration of all the subcarriers, the distortion of subcarrier orthogonality is avoided.

The *v*th subcarrier component is given as follows.(3)$$\begin{array}{c} Y\left(v\right)=\sum_{m=1}^{N}y\left(m\right)\exp\left(-j2\pi \frac{mv}{N}\right),\\ =\sum_{k=1}^{K}\gamma_{k}\sum_{m=1}^{N}\exp\left(j2\pi \frac{n_{k}m+m\varepsilon_{k}-mv}{N}\right)+\sum_{m=1}^{N}n_{0}\left(m\right)\exp\left(-j2\pi \frac{mv}{N}\right) \end{array}$$where $\gamma_{k}$ is a CSI of the wireless channel between *k*th user and FC and it is defined as follows.(4)$$\begin{array}{c} \gamma_{k}=\sum_{l=1}^{L}h_{k,l}\exp\left(j2\pi \frac{-n_{k}\tau_{k,l}}{N}\right) \end{array}$$

From the conversion table between the subcarrier number and the sensing results, FC can recognize all the sensing results and then can also evaluate the median and the outlier of all the sensing results at once.

If the second term of Equation (Equation 3), which is the noise component, is ignored, the subcarrier component in $v\neq n_{k}$ appears and it is an ICI. For simple explanation, the spectrum of subcarrier in $v=n_{k}$ and $v\neq n_{k}$ are the main lobe and side-lobe, respectively. The ICI components mask the subcarrier components sent by the other sensor and generate the image subcarrier that causes the false subcarrier detection by FC [6].

### 3.2. Sensing Results Separation

The insertion scheme of ID for specifying the information source, i.e., sensor, has been considered. For example, the particular time hopping sequence for each sensor [24] and the data tracking with the feature amount of sensing result such as time continuity are available for specifying each sensor from the received sensing results [6]. However, the former scheme requires more time with the resources for informing ID and thus the signaling overhead of inserting ID is significant. In addition, it also requires highly accurate timing synchronization. In the latter scheme, when the sensing results are near together, the tracking misunderstands the different sensor because the particular amount of sensing results is similar. Once the misunderstanding occurs, the sensing result with wrong sensor is tracked. The error event is referred to as an error tracking [7]. Error tracking is a serious problem because it significantly degrades the strictness of the data recognized by FC.

## 4. Proposed Sensing Results Separation

The basic principle of the proposed sensing result separation is the same as data tracking [6] and [7]. In FC, the data tracking separates the sensing results with common sensor node from all the recognized sensing results.

### 4.1. Sensor Track based on Fractional CFO

FC assigns the particular fractional CFO to each sensor for using the ID of the sensor, where the fractional CFO is defined as the smaller frequency shift than the frequency minimum duration of DFT in FC. We distinguish the increment and decrement of frequency shift and thus CFO includes the sign (plus and minus). FC can specify the sensor node from the estimated CFO. Since the value of CFO is smaller than the frequency minimum duration of DFT, the consumption of frequency resource by inserting ID is limited. However, the ICI caused by CFO occurs and the compensation to ICI is required.

In the proposed separation, the two protocols are selected in accordance with the following two situations. In the first situation, each sensor takes different sensing result and thus the subcarrier sent by each user is not interfered. In the second situation, certain sensors take the same sensing result and thus the subcarrier sent by them are interfered, where taking the same sensing result is referred to as the duplication of sensing results. Before FC selects the protocols, it detects the duplication of the sensing result. Figure 2 shows the flow of proposed data separation. It is composed of three parts, detector or duplication of sensing result, the sensing results separation without duplication of sensing result, and the sensing results separation with it. In the following section, we explain each protocol in detail.

### 4.2. Sensing Results Separation without Duplication of Sensing Results

The protocol of data separation is composed of three parts, the estimation of CSI and CFO, the construction of ICI replica and the iterative ICI cancellation, and the data tracking with sensing results, CSI, and CFO.

#### 4.2.1. Spectrum Detection and Estimating CSI and CFO

Since each sensor sends the subcarrier with different frequency to FC, at most one main lobe of subcarrier exists in each DFT point. Without loss of generality, we consider FC detects the subcarrier sent by *k*th sensor. From Equation (Equation 3), the *v*th subcarrier component is given as follows.(5)$$\begin{array}{c} Y_{k}\left(v\right)=\gamma_{k}\sum_{m=1}^{N}\exp\left(j2\pi \frac{n_{k}m+m\varepsilon_{k}-mv}{N}\right). \end{array}$$

For simple explanation, the noise component is ignored. The proposed technique evaluates the phase rotation speed for estimating the frequency offset [25]. FC detects the signal during the window size of DFT for two or more times in a row. For simple explanation, this paper considers twice detections.

We consider the subcarrier components detected by the first DFT are given by Equation (Equation 5). After first DFT detection, the *v*th subcarrier component in the second DFT detection is given as follows.(6)$$\begin{array}{c} Y_{k}^{\prime }\left(v\right)=\gamma_{k}\sum_{m=1}^{N}\exp\left(j2\pi \frac{n_{k}\left(m+N\right)+\left(m+N\right)\varepsilon_{k}-\left(m+N\right)v}{N}\right), \end{array}$$where $\gamma_{k}$ and $\varepsilon_{k}$ are constant during two times detections of DFT. Since $n_{k}$ and *v* are integer, the following equation is set.(7)$$\begin{array}{c} Y_{k}^{\prime }\left(v\right)=\gamma_{k}\sum_{m=1}^{N}\exp\left(j2\pi \frac{n_{k}m+\left(m+N\right)\varepsilon_{k}-mv}{N}\right). \end{array}$$

Note that $\gamma_{k}$ does not depend on the subcarrier number *v* but it is constant for the frequency domain. Therefore, as far as |ε|<0.5 is set, the spectrum selected in terms of maximal spectrum power is equal to the main lobe of the subcarrier. As a result, the $v=n_{k}$ spectrum components in first and second DFT detections are given as follows.(8)$$\begin{array}{c} Y_{k}\left(v=n_{k}\right)=\sum_{m=1}^{N}\exp\left(j2\pi \frac{m\varepsilon_{k}}{N}\right)\gamma_{k}\\ Y_{k}^{\prime }\left(v=n_{k}\right)=\sum_{m=1}^{N}\exp\left(j2\pi \frac{\left(m+N\right)\varepsilon_{k}}{N}\right)\gamma_{k}\\ =\exp\left(j2\pi \varepsilon_{k}\right)Y\left(v=n_{k}\right). \end{array}$$

Finally, the CFO is estimated as(9)$$\begin{array}{c} \varepsilon_{k}=\log\left(\frac{Y_{k}^{\prime }\left(v=n_{k}\right)}{Y_{k}\left(v=n_{k}\right)}\right)\frac{1}{j2\pi }. \end{array}$$

In addition, the CSI is estimated as(10)$$\begin{array}{c} \gamma_{k}=\frac{Y_{k}\left(v=n_{k}\right)}{\sum_{m=1}^{N}\exp\left(j2\pi \frac{m\varepsilon_{k}}{N}\right)}. \end{array}$$

#### 4.2.2. Construction of ICI Replica and Process of Interference Cancellation

In Equation (Equation 5), the $v\neq n_{k}$ spectrum components, $Y(v\neq n_{k})$, are ICI. Note that the CSI, $\gamma_{k}$, is constant regardless of subcarrier number, *v*. Therefore, the CSI estimated in main lobe is used for constructing ICI replica. After the CFO and the CSI are estimated using Equations (Equation 9) and (Equation 10), respectively, the ICI replica is constructed using Equation (Equation 5). FC can substitute the constructed ICI replica from the detected frequency spectrum given by Equation (Equation 3) and thus it can remove the ICI components.

After ICI cancellation, the spectrum with maximal power is selected from the detected frequency spectrum again. As a result, the main lobe of the subsequent subcarrier can be detected. After that, the estimation of CSI and CFO, the construction of ICI replica, and the cancellation of ICI are performed repeatedly, until the subcarriers sent by all the sensor nodes are detected. In the proposed technique, for detection of the main lobe of the subcarrier, ICI is removed. Therefore, the masking of ICI to the other subcarrier and the false detection of subcarrier are avoidable. Note that in PhyC-SN the number of detected subcarriers is equal to the number of sensor nodes as far as the duplication of sensing results does not occur.

#### 4.2.3. Data Tracking for Sensing Result Separation

FC can obtain the data set composed of sensing result, which is equal to the frequency number in subcarrier, CSI, and CFO owing to the spectrum detection and the estimation of CFO and CSI. The detected data set is considered to be the particular indicator for specifying the sensor node. FC assigns the label of sensor node to each data set for the sensing result separation into several sensing results of each sensor node. When all the sensors send subcarriers to FC in *t*th time slots, the temporal number, i(∈1,2,…,K), is assigned to each data set, where the sensing result, the CSI, and the CFO in the *i*th data set are $n_{i},\gamma_{i},\varepsilon_{i}$, respectively. Figure 3 shows the image of label assignment.

When all the sensors send the subcarriers to FC in the (t+1)th time slot, the temporal number, j(∈1,2,…,K), is assigned to each data set, where the sensing result, the CSI, and the CFO in the *j*th data set are $n_{j},\gamma_{j},\varepsilon_{j}$, respectively. Therefore, the cost function required for the transition from the *i*th data set to the *j*th data set is defined as follows.(11)$$\begin{array}{c} f\left(i,j\right)=\frac{{\left(n_{i}-n_{j}\right)}^{2}}{A}+\frac{{\left|\gamma_{i}-\gamma_{j}\right|}^{2}}{B}+\frac{{\left(\varepsilon_{i}-\varepsilon_{j}\right)}^{2}}{C}, \end{array}$$where A,B, and *C* are the normalization coefficients.

If the sensing result, CSI, and CFO exhibit time continuity, the transition with minimum cost of Equation (Equation 11) is the most powerful. We can consider that multiple data sets in the *t*th time slot are not connected to a data set in the (t+1)th time slot because the duplication of sensing results does not occur. The number of transition patterns is K!, where the *s*th class (∈1,2,…,S=K!) of valid transition pattern is $u_{s}$. As a result, the label assignment of sensor node is considered to be the following optimization problem.(12)$$\begin{array}{c} s^{*}=\arg\min_{s}\sum_{\left\{i,j\right\}\in u_{s}}f\left(i,j\right), \end{array}$$where $s^{*}$ is the class of transition pattern with the minimum cost. For solving this optimization problem, the brute-force protocol is considered. It is that the costs of all the transition patterns are evaluated and then the best transition pattern, which archives the minimum cost, is selected. As *K* becomes larger, the number of transition patterns is dramatically increased. For relaxing the solution with low complexity, the sphere decoding is available [26]. In the sphere decoding, if the cost of transition is over a certain value, the transition pattern including it is out of candidates.

### 4.3. Sensing Results Separation with Duplication of Sensing Results

#### 4.3.1. Detecting Duplication of Sensing Results

We assume certain sensors, or all the sensors take common sensing result. Figure 4 shows the image of the data set in the *t*th and (t+1)th time slots. We define the sensors taking the sensing result in the *j*th data set as the sensor group of $p_{j}\in I^{D_{j}}$, where I is the integer field and $D_{j}$ is the number of sensors taking the sensing data in *j*th data set.

In the (t+1)th time slot, the $n_{j}$th spectrum detected via DFT is given as follows(13)$$\begin{array}{c} Y\left(n_{j}\right)=\sum_{k\in p_{j}}\gamma_{k}\sum_{m=1}^{N}\exp\left(j2\pi \frac{m\varepsilon_{k}}{N}\right). \end{array}$$The noise components are also ignored. Since the subcarrier sent by each sensor is interfered with each other, the CSI and the CFO in each sensor node is hardly estimated. In the proposed separation, regardless of the duplication of sensing results, FC selects the spectrum with the maximal power and estimates the CSI and the CFO from the selected spectrum. Even under the duplication of the sensing results, it hardly occurs that the power of side-lobe of subcarrier is larger than that of the main lobe of subcarrier as large as the absolute value of CFO is smaller than 0.5. Therefore, it is reasonable assumption that the selected spectrum with maximal power is equal to the main lobe of the subcarrier even under the duplication of sensing results. From the *j*th subcarrier, the CFO, $\hat{\varepsilon_{j}}$, is estimated in accordance with Equation (Equation 9). Note that the estimated CFO, $\hat{\varepsilon_{j}}$, is different from the individual CFO of each sensor because the mutual interference among subcarriers is exceedingly large. The CSI of the *j*th subcarrier is estimated as(14)$$\begin{array}{c} \hat{\gamma_{j}}=\frac{Y(n_{j})}{\sum_{m=1}^{N}\exp\left(j2\pi \frac{m\hat{\varepsilon_{j}}}{N}\right)}. \end{array}$$As a result, the constructed ICI replica is given as(15)$$\begin{array}{c} \hat{Y}\left(n_{j}\right)=\frac{Y(n_{j})}{\sum_{m=1}^{N}\exp\left(j2\pi \frac{m\hat{\varepsilon }}{N}\right)}\sum_{m=1}^{N}\exp\left(j2\pi \frac{n_{k}m+m\hat{\varepsilon }-mv}{N}\right). \end{array}$$It is different from the original ICI given by Equation (Equation 3). Therefore, the effect of suppressing ICI is limited. However, the main lobe of the constructed replica is matched to the original one because the detected spectrum, $\hat{Y}\left(n_{j}\right)$ is the same as the original one $Y(n_{j})$. Therefore, the main lobe of the subcarrier can be removed. In the proposed separation, the scheme for counting the types of subcarriers is proposed, where a type of subcarrier is the subcarrier selected by one or more sensor nodes.

We define the protocol, which is the selection of subcarrier in terms of maximal power criterion, the estimation of CSI and CFO, and the cancellation by constructed ICI replica as the iterative protocol. After the iterative protocol, FC can detect the subcarrier with power exceeding a certain threshold. If it is true, the iterative protocol is performed again and then comparison between the power of subcarrier and the certain threshold is conducted. Otherwise, it is stopped. If the number of constructed ICI replicas during the iterative protocols is equal to the number of sensors, *K*, the duplication of sensing results does not occur. Otherwise, it occurs.

If the CFO is so large that the power of ICI is large, the residual ICI is still large even after ICI cancellation. Especially, if the duplication of sensing results occurs, it is large because the mitigation of ICI is limited. Therefore, the CFO should be so small that ICI is not so large.

#### 4.3.2. Data Tracking Based on Vector of Detected Subcarrier

As FC can detect the duplication of the sensing result by counting the type of subcarriers, the estimated CSI and CFO are not available for data tracking because the estimated CSI and CFO are different from those of each sensor. In the proposed separation, the data tracking based on the vector of subcarrier is proposed.

We assume high time correlation of CSI and CFO during two time slots at least. We consider the duplication of sensing results does not occur in *t*th time slot and occurs in (t+1) time slot. From Equation (Equation 13), the spectrum of subcarrier, which certain sensors commonly select in (t+1)th time slot is the linear combination among the spectrum of the subcarriers, which these select in *t*th time slot. Therefore, we consider the connection of the data set between the two time slots. In *t*th time slot, the $i\in \left\{1,2,\ldots K\right\}$th label is assigned to the data set including sensing result, CSI, and CFO. In (t+1)th time slot, $j\in \left\{1,2,\ldots ,K^{\prime }\right\},K^{\prime }<K$ is assigned to the data set, where $K^{\prime }$ is the types of detected subcarriers in (t+1)th time slot. The problem of connecting data set is similar to that without duplication of sensing results but the data set in (t+1) time slot can be connected to multiple data sets in *t*th time slot. As a result, the total number of connection patterns, $S^{\prime }$, is given as follows.(16)$$\begin{array}{c} S^{\prime }={K^{\prime }}^{K}-\sum_{k=1}^{K^{\prime }-1}{\;}_{K^{\prime }}C_{k}k^{K}, \end{array}$$where ${}_{a}C_{b}$ is the number of combinations for *b* samples selected from *a* samples. ${u^{\prime }}_{s}$ is defined as the $s(\in 1,2,\ldots ,S^{\prime })$th class of connection combination.

In (t+1)th time slot, the group of sensors, which select the *j*th subcarrier is defined as $p_{j}$. The *j*th subcarrier is reconstructed using subcarrier components in *t*th time slots and it is given as follows.(17)$$\begin{array}{c} \Lambda_{j}=\sum_{i\in p_{j}}\gamma_{i}\sum_{m=1}^{N}\exp\left(j2\pi \frac{m\varepsilon_{i}}{N}\right). \end{array}$$$\Lambda_{j}$ is referred to as the prediction of *j*th subcarrier. The spectrum detected in *t*th time slot is switched into $\Lambda_{j}$ and then the new spectrum is given as follows.(18)$$\begin{array}{c} Y\left(n_{i}\right)=\Lambda_{j}\left(i\in p_{i}\right). \end{array}$$

As a result, the following cost function for the transition from *j*th data set to *i*th data set is given as(19)$$\begin{array}{c} g\left(i,j\right)={\left(\frac{n_{i}-n_{j}}{D}\right)}^{2}+{\left|\frac{Y\left(n_{i}\right)-Y\left(n_{j}\right)}{E}\right|}^{2}. \end{array}$$

Since the subcarrier component is composed of the CSI and CFO, which are particular amount for each sensor, it is also particular amount. If CSI and CFO exhibit time continuity, we can consider subcarrier component also has time continuity. Therefore, the transition with minimum cost of Equation (Equation 19) is a most powerful. Therefore, selecting the suitable transition pattern is considered to be the following optimization problem.(20)$$\begin{array}{c} s^{*}=\arg\min_{s}\sum_{\left\{i,j\right\}\in u_{s}^{\prime }}g\left(i,j\right). \end{array}$$After recovering the above optimization problem, the data separation for each sensor is achieved.

After deciding the connection between the data sets, the data set in (t+1)th time slot connected to multiple data set in *t*th time slot requires the update of CSI and CFO because the individual CSI and CFO are not obtained due to large interference. In the proposed separation, the data set connected to multiple data sets is divided into the individual data set using that in *t*th time slot. Figure 5 shows the image of dividing the data set in (t+1)th time slots. Although the sensing results are common for the divided data set, the CFO and the CSI of the data set in (t+1)th time slot are overwritten by the CFO and the CSI of data in *t*th time slot. Therefore, the number of data sets in (t+1) slots is increased, and it is equal to *K*. Owing to this, the subsequent data tracking from (t+1)th time slot to (t+2)th time slot can use the individual data set of each sensor and thus the protocol of data separation is continued.

## 5. Numerical Results

### 5.1. Simulation Results

The number of DFT points, *N*, is 256. A trial is composed of 30 times transmission of sensing results. In computer simulation, 1000 trials are performed for evaluating the accuracy of data collection. The CFO of the *k*th sensor in (t+1)th time slot is defined as the following equation using that in *t*th time slot.(21)$$\begin{array}{c} \varepsilon_{k}\left(t+1\right)=\rho_{\varepsilon }\varepsilon_{k}\left(t\right)-\sqrt{1-{\rho_{\varepsilon }}^{2}}\varepsilon_{0} \end{array}$$
(22)$$\begin{array}{c} \varepsilon_{k}\left(1\right)=\frac{2\varepsilon_{M}\left(k-1\right)}{K-1}-\varepsilon_{M},\;\;\;t\geq 1, \end{array}$$where $\varepsilon_{0}$ is a uniform random variable with $\left[-\varepsilon_{M},\varepsilon_{M}\right]$ and $\varepsilon_{M}$ being the maximal value of CFO. $\rho_{\varepsilon }$ is the time correlation of CFO. $\varepsilon_{k}\left(1\right)$ is the initial value of CFO of *k*th user. For deciding the initial CFO, the range from $-\varepsilon_{M}$ to $\varepsilon_{M}$ is uniformly divided into *K* values and each value of CFO is assigned to each user.

The CSI of *k*th sensor in (t+1)th time slot is decided as the following equation by that in *t*th time slot.(23)$$\begin{array}{c} \gamma \left(t+1\right)=\rho_{\gamma }\gamma \left(t\right)+\sqrt{1-{\rho_{\gamma }}^{2}}\gamma_{0}, \end{array}$$where $\rho_{\gamma }$ is the time correlation of CSI and $\gamma_{0}$ is the random variable with independency for each user and each time slot. We assume the two paths with equivalent delay profile whose path duration is 1 sample. First path and second path are modeled by a rice fading with 10 dB rice factor and a Rayleigh fading, respectively.

The primary objective of the computer simulation is explaining the effect of ICI cancellation and the accuracy of sensing result separation. In this paper, the evaluation of robustness with respect to the noise component is future work and thus the simulation assumes the signal power to be sufficiently large to avoid the false alarm, which means FC wrongly recognizes the noise component as the subcarrier sent by sensor.

In the proposed technique, the threshold for confirming the existence of subcarrier is 1/10 smaller than the average signal power.

The sensing result is modeled by random walk and it is defined as(24)$$\begin{array}{c} n_{k}\left(t=0\right)=\left\lfloor \frac{N}{\left(10(K+1)\right)}\right\rfloor k+\left\lfloor \frac{N}{4}\right\rfloor \end{array}$$
(25)$$\begin{array}{c} n_{k}\left(t+1\right)=n_{k}\left(t\right)+\left[\beta \right], \end{array}$$where ⌊·⌋ is floor function and [·] is the rounding off function. β is a uniform random variable with [−3,3]. In the initial time slot, the different sensing result is assigned to each user. For data tracking, in the cost functions given by Equation (Equation 11) and Equation (Equation 19), the normalization factors of sensing result, CSI, CFO, and received spectrum are the median of total subcarrier numbers, the average power of CSI, the maximal CFO, $\varepsilon_{M}$, and the average power of received spectrum, respectively.

For evaluating the accuracy of sensing results separation, we define a root mean square error (RMSE) as follows.(26)$$\begin{array}{c} \mathrm{RMSE}=\sqrt{E\left[\frac{{\left(\bar{n}-\hat{n}\right)}^{2}}{{\hat{n}}^{2}}\right]}, \end{array}$$where $\bar{n}$ is the sensing result detected by FC and $\hat{n}$ is the true sensing result. Root mean square error (RMSE) is averaged for all the trials and all the sensors and thus the average RMSE is derived. Figure 6 shows the snap shot of the separation results. The ground truth and the tracked result are the original sensing result and the result separated by tracking, respectively. We can observe in RMSE = 0.0123, the separation error is limited but if RMSE exceeds 0.0123, the burst separation errors occur, and these are error tracking. Therefore, our required RMSE is approximately 0.01.

Figure 7 shows the performance between maximal CFO $\varepsilon_{M}$ and a RMSE, where the number of sensors, *K*, is 5, the time correlations of CSI $\rho_{\gamma }$ and CFO $\rho_{\varepsilon }$ are 0.999. “Energy” (“Energy is the data separation proposed in [6]) is a conventional data separation. In “Energy”, the subcarriers are detected in terms of the top *K* spectrum power and the cost function of the data separation is composed of sensing results for tracking the time continuity of sensing results. Both of “Cancel w/o Adaptation” (“Cancel w/o Adaptation” is the data separation proposed in [27]) and “Cancel w/ Adaptation” use ICI cancellation for detecting the subcarriers. In both, the cost function of data separation is composed of sensing results, CSI and CFO and it is given by Equation (Equation 11). In “Cancel w/ adaptation”, while detecting the duplication of the sensing results, the data tracking is switched into that composed of sensing results and reconstructed spectrum which is given by Equation (Equation 19).

From this figure, “Cancel w/ Adaptation”, which is proposed separation, achieves the minimum RMSE and thus results in highly accurate data separation. As the maximal CFO becomes larger, the RMSE of “Cancel w/ Adaptation” is degraded. The effect of ICI cancellation under the duplication of sensing results is limited. As CFO becomes larger, the power of ICI becomes larger and thus the residual ICI after ICI cancellation becomes larger. Therefore, the wrong recognition that the residual ICI is considered to be the subcarrier sent by the sensor occurs. As a result, RMSE is degraded. From these results, as the maximal CFO is smaller than 0.05, the average RMSE under 0.01 is achieved.

Figure 8 shows the performance between the time correlation of CSI and CFO, $\rho_{\gamma },\rho_{\varepsilon }$, and the average RMSE, where the time correlation of CSI is equal to that of CFO, $\rho_{\gamma }=\rho_{\varepsilon }$. The number of sensor nodes, *K*, is 5. The maximal CFO, $\varepsilon_{M}$, is 0.05. In “Cancel w/o Adaptation”, the improvement of RMSE is limited. This is because the duplication of sensing results causes the misunderstanding of sensor node and thus the error tracking causes the degradation of RMSE. “Cancel w/ Adaptation” is the best performance. As $\rho_{\gamma },\rho_{\varepsilon }$ becomes larger, the RMSE of “Cancel w/ Adaptation” is improved. Owing to the high time continuity of CSI and CFO, the accuracy of data tracking is improved. In addition, the tracking with received spectrum aids in avoiding the error tracking.

Figure 9 shows the performance between the number of sensor nodes, *K*, and average RMSE. The maximal CFO, $\varepsilon_{M}$, is 0.05 and the time correlations of CSI and CFO, $\rho_{\varepsilon },\rho_{\gamma }$ are 0.999. From this figure, “Cancellation w/ Adaptation” achieves smallest RMSE. As the number of sensors becomes larger, the RMSE is degraded. This is because the occurrence probability of duplication of sensing results becomes larger and the power residual ICI by ICI cancellation also becomes larger.

In this subsection, the sensing results obtained by the actual sensors are used for the sensing result in computer simulation. The purpose of performance evaluation is clarifying the accuracy of sensing result separation by proposed scheme in applying to the actual sensing results. We use two temperature sensing results. First one is as follows. Figure 10 and Figure 11 show the images of experimental evaluation and the obtained sensing results, respectively. In this experimental evaluation, the temperature sensor is used and the number of sensors, *K*, is 5. Every 1 minute, each sensor evaluates the temperature from 4 am to 1 pm in the last ten days of April. Since an automatic air conditioner is stopped, the cause of changing temperature is a climate shift, such as sunshine condition, wind condition, and so on. Figure 12 shows the obtained temperature results in the second situation and this test sensing results are given in [6]. Every 1 second, each sensor evaluates the temperature during 500 s in the end of November. We start working a heater after 150 s from start time. The other detail conditions are shown in [6]. In the first result, the impact of sunny and wind conditions to sensor is mild and the sensing results are almost commonly changed. The time correlation and the correlation among sensors are high. In the second result, a heater makes the temperature enlarged. Especially, the nearer to the heater the sensor is, the more significant the increment of temperature is. Therefore, the difference of temperature among sensor is larger in the second result than that in the first result.

A wireless communication is constructed by computer simulation. The wireless propagation model is the same as the computer simulation discussed in previous subsection. The maximal CFO, $\varepsilon_{M}$, is 0.05 and the time correlation of CFO and CSI, $\rho_{\gamma }=\rho_{\varepsilon }$, is 0.999999. In only the proposed technique, the RMSE is evaluated under $\rho_{\gamma }=\rho_{\varepsilon }=0.99999$.

### 5.2. Performance Evaluation Based on Actual Sensing Results

Figure 13 shows the cumulative distribution function (CDF) of RMSE in the sensing results of Figure 11. From this figure, “Cancel w/ Adaptation” achieves the minimum RMSE among all the results. As $\rho_{\gamma }$ and $\rho_{\varepsilon }$ are changed from 0.999999 to 0.99999, the RMSE is degraded. The temperature sensor has the high correlation among sensors. Therefore, each sensing result takes the similar value. As a result, in the detection by FC, the duplication of sensing results frequency occurs. In the proposed technique, the data set is updated from the past data set. If the updating of the data set continues for long time slots, the mismatch of the detected CSI, CFO, frequency spectrum between detecting data set and past one becomes larger. Therefore, the high time correlation of CSI and CFO are required for achieving smaller RMSE.

Figure 14 shows the CDF of RMSE in the sensing results of Figure 12. We can see that all the performances of this figure are better than those of Figure 13. This is because the duplication of sensing results occurs in the fewer times. In addition, the difference of “Cancel w/ Adaptationff between $\rho_{\varepsilon }=\rho_{\gamma }=0.99999$ and $\rho_{\varepsilon }=\rho_{\gamma }=0.9999$ becomes smaller. This reason is as follow. Although the time correlation of CSI and CFO becomes low, the better data tracking is achieved owing to the highly time continuity of sensing results. From the evaluations with using actual sensing results, our proposed technique achieves the better data separation with the lower correlation of sensing results among sensors and the more highly time correlation of sensing results.

## 6. Conclusions

This paper proposed the insertion technique of ID with CFO to each transmit signal and the sensing data separation with the ICI cancellation and the data-tracking technique for a physical wireless parameter conversion sensor network (PhyC-SN). In the proposed technique, the FC assigned the fractional CFO to each user. The assignment of fraction CFO is useful for specifying the sensor node. This paper also proposes the construction of ICI replica in a narrow band wireless communication system. For it, the estimation of the CSI and the CFO from the received signals is considered. After ICI cancellation, the tracking technique using the detected data set including the sensing result, CFO, and CSI is proposed. It can recover the problem of PhyC-SN that does not specify the sensor from the obtained sensing results. If certain sensors take the common sensing results, the duplication of subcarriers occurs. For compensating it, this paper proposed the detection scheme for duplication of sensing results and the data tracking based on the received spectrum. Based on computer simulation, the advantage of the proposed technique in terms of accuracy of data separation is elucidated.

However, there are two disadvantages of proposed separation and ID insertion. If the data duplication continuously and frequently occurs, the accuracy of separation is degraded because it continuously uses the past estimated CSI and CFO and thus the difference between the past estimated CSI and CFO and the practical ones becomes larger. It is one of disadvantages. In second disadvantages, the process of adding the fractional frequency offset to carrier signal is complicated for the sensor. Recovering these disadvantages is important future works.

In practical wireless environment, the CSI is fluctuated [8] and the CFO is also done [9]. The proposed technique is composed of three kinds of costs for tracking, sensing results, CSI, and CFO. Therefore, the experimental evaluation of data transmission is also important future work for clarifying the accuracy of proposed data separation under the more practical wireless environment.

## Figures and Tables

**Figure 1 sensors-19-00767-f001:**
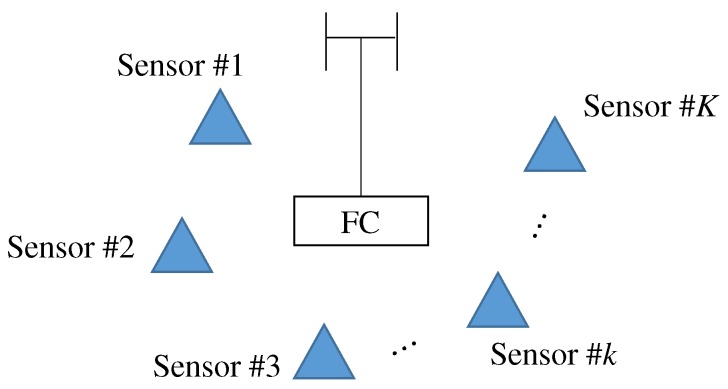
Overview of wireless sensor networks.

**Figure 2 sensors-19-00767-f002:**
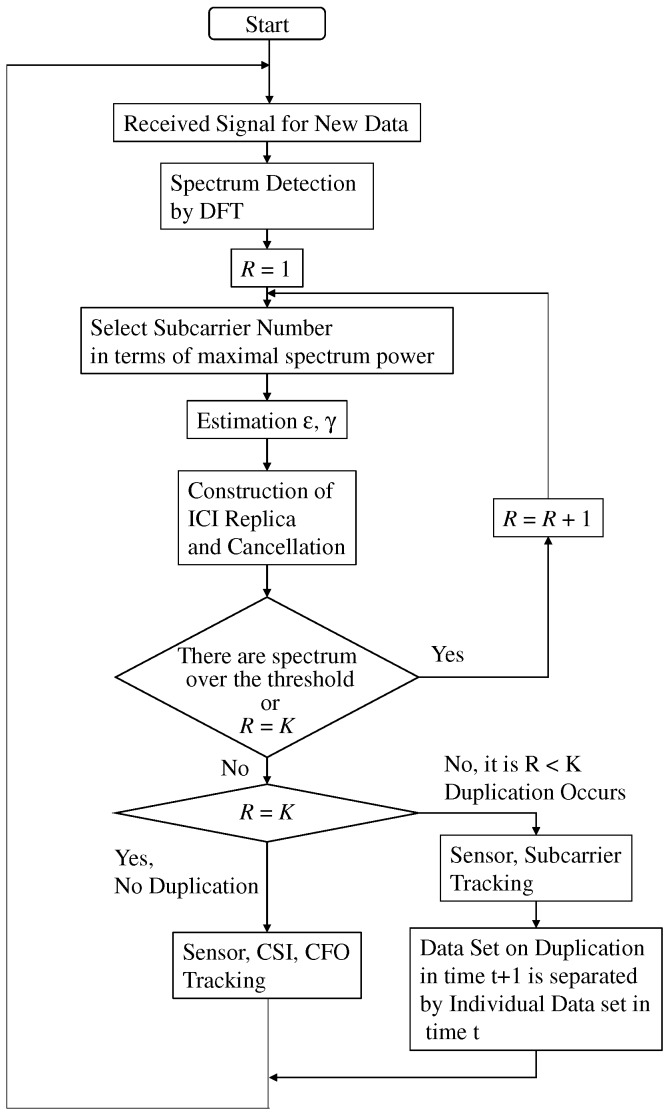
Flow chart of proposed ICI cancellation and data tracking.

**Figure 3 sensors-19-00767-f003:**
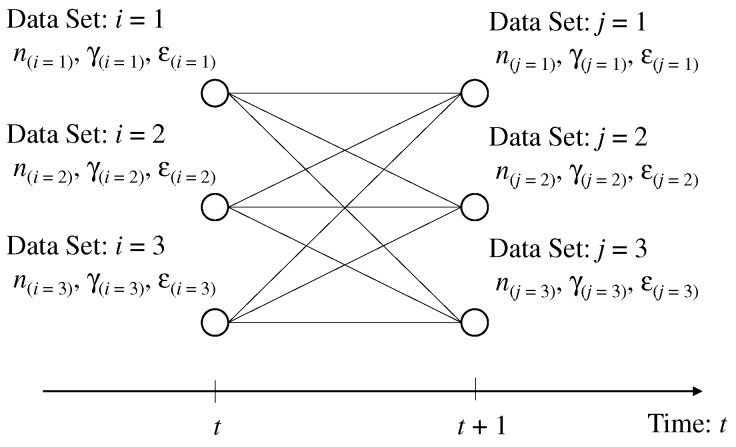
Example of labeling problem for data separation without the duplication of sensing results.

**Figure 4 sensors-19-00767-f004:**
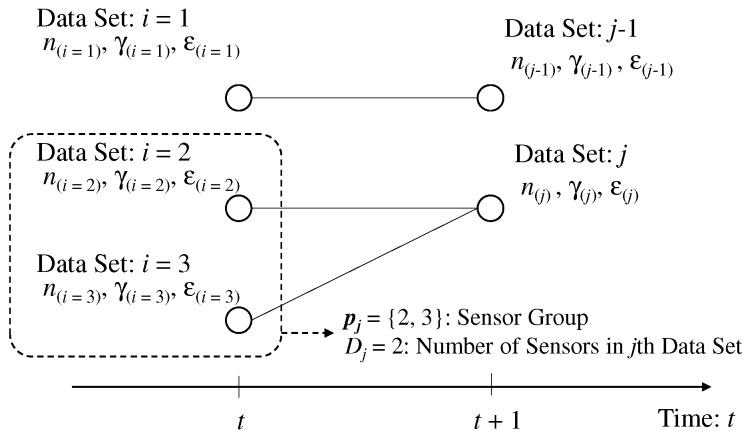
Example of labeling problem for data separation with the duplication of sensing results.

**Figure 5 sensors-19-00767-f005:**
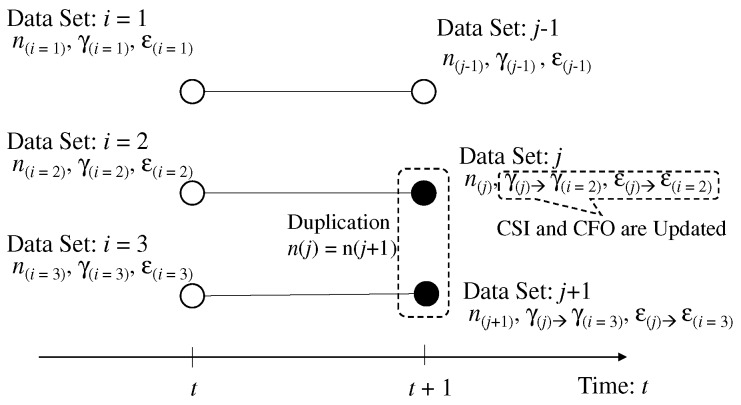
Updating of data set after labeling with duplication of sensing results.

**Figure 6 sensors-19-00767-f006:**
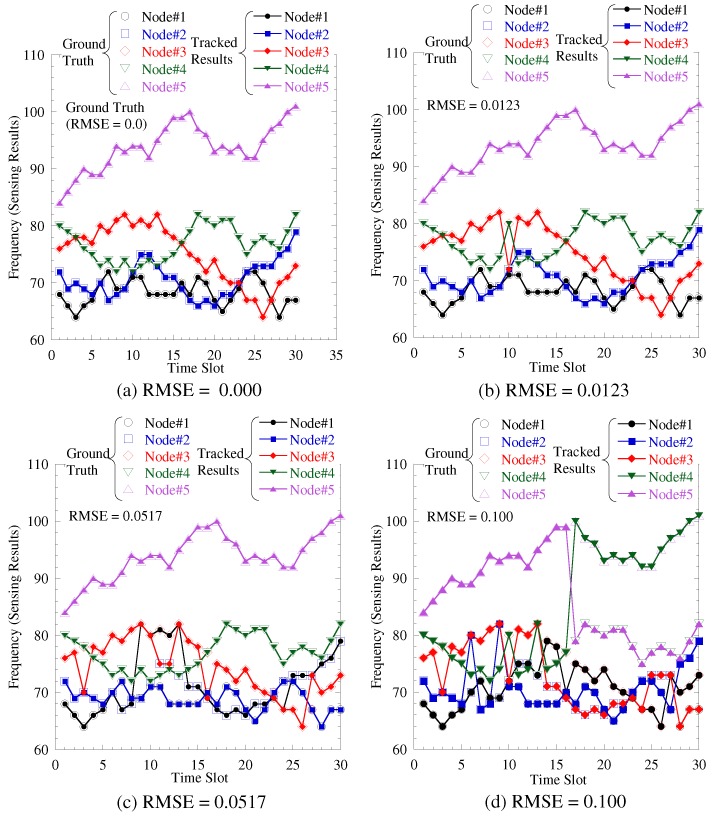
Snap shot of separation result under various RMSE.

**Figure 7 sensors-19-00767-f007:**
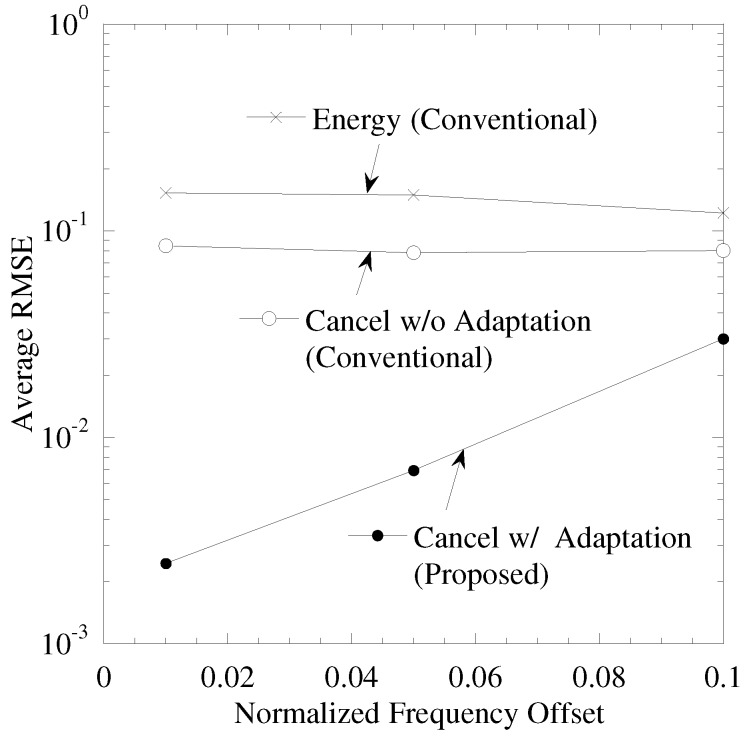
Performance between maximal frequency offset and RMSE.

**Figure 8 sensors-19-00767-f008:**
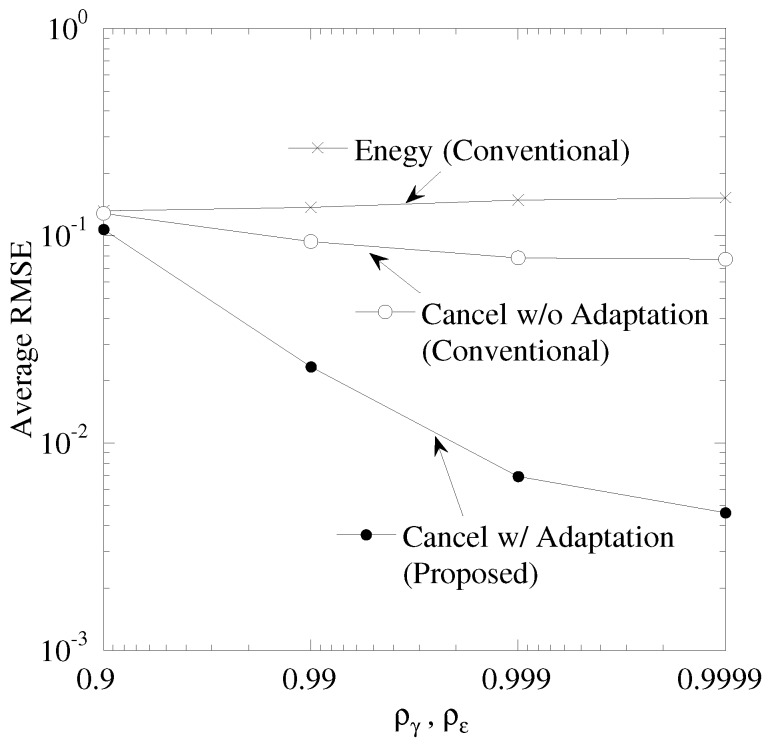
Performance between $\rho_{\gamma }=\rho_{\varepsilon }$ and RMSE.

**Figure 9 sensors-19-00767-f009:**
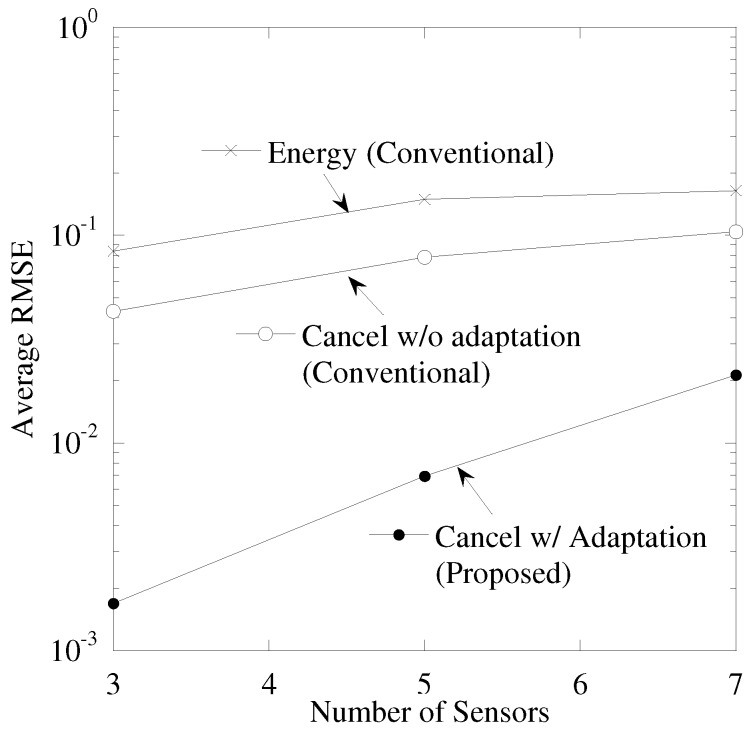
Performance between number of sensors and RMSE.

**Figure 10 sensors-19-00767-f010:**
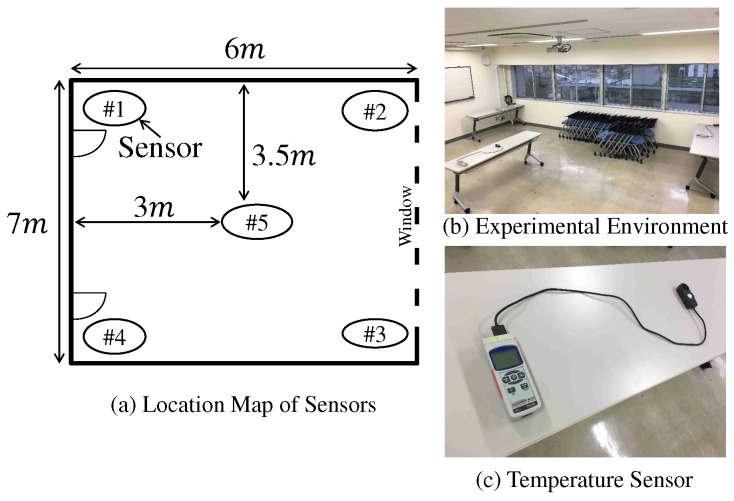
Overview of experimental evaluation.

**Figure 11 sensors-19-00767-f011:**
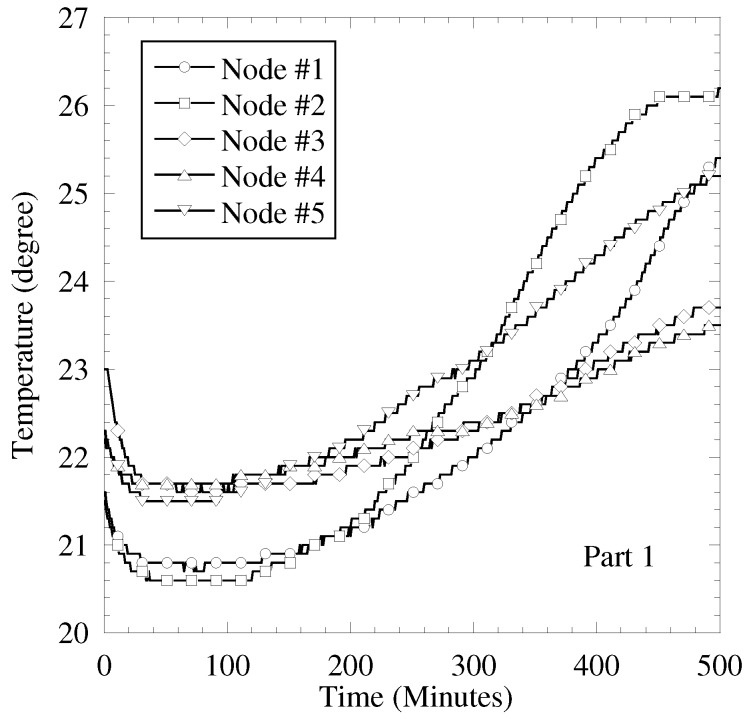
Results of temperature sensor in Part 1.

**Figure 12 sensors-19-00767-f012:**
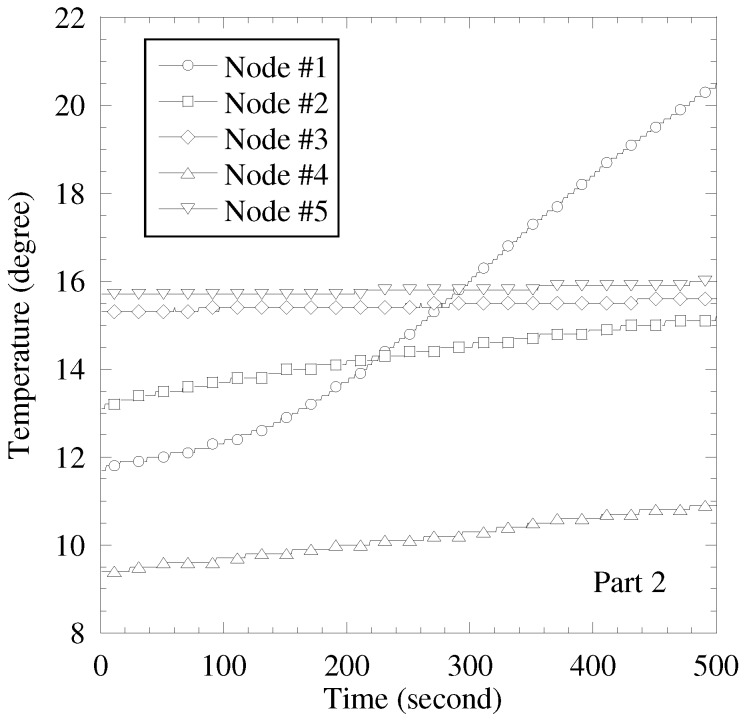
Results of temperature sensor in Part 2.

**Figure 13 sensors-19-00767-f013:**
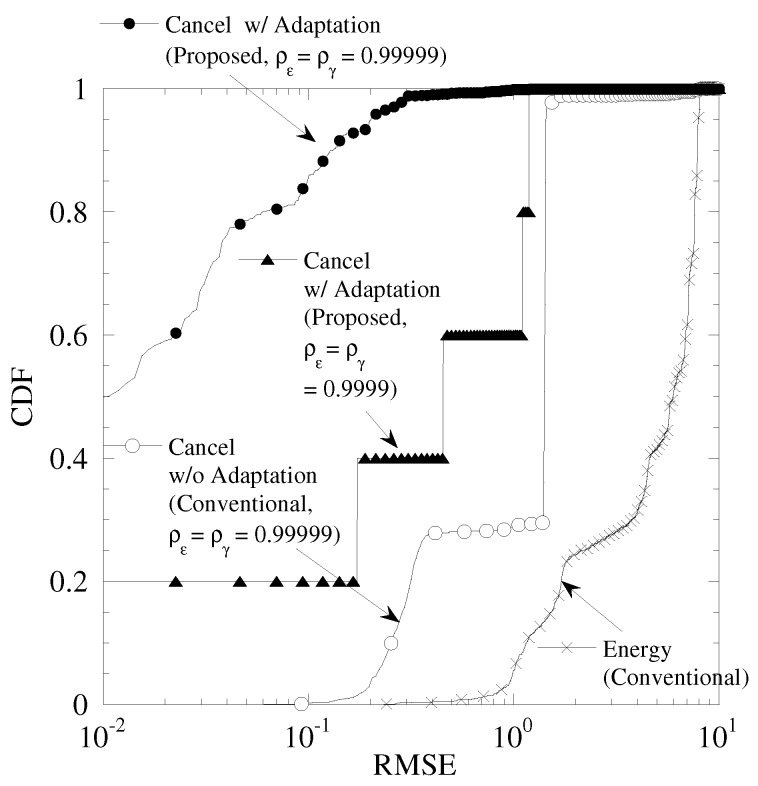
CDF of RMSE in temperature sensor of Part 1.

**Figure 14 sensors-19-00767-f014:**
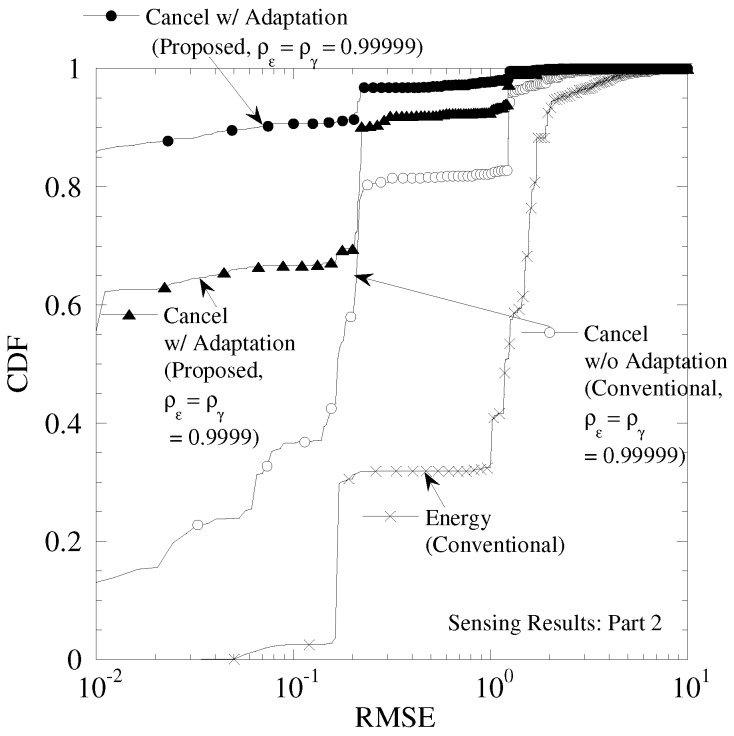
CDF of RMSE in temperature sensor of Part 2.

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
