# Peer review of "ID Insertion and Data Tracking with Frequency Offset for Physical Wireless Parameter Conversion Sensor Networksâ€"

_sensors, 2019, doi:10.3390/s19040767_

Round 1
Reviewer 1 Report
The experimental evaluation paragraph cannot be publish without any major modifications.
First of all the experiment is not clearly detailled:
- what is the temperature change?
- figure 13 is a simulation result not an experimental one.
Because a part of the theoretical results have been yet published, the experimental part should be the added value of the paper. So the paper cannot be publish as soon as this experiment is fully realized and well explained. Experimental transmission must be realized.
Actually the paper don't provide any added value compare to the paper published in IEEE Wireless Communications and Networking Conference in 2017
and entitled :
Data Tracking Using Frequency Offset and SIC for Physical Wireless Conversion Sensor Networks
7 Author(s)
Takehiro Sakai; Osamu Takyu; Keiichiro Shirai; Mai Ohta; Takeo Fujii; Fumihito Sasamori; Shiro Handa
The paper needs major revisions. Complete experiments should be made to validate the theoretical results.
Author Response
Dear Dr. Reviewer 1
Thank you very much for spending your precious time to review our paper.
Thanks to your important comments, the revised paper is much better than the original one. Please, re-review our paper, again. We would like to show the answer to your comments.
The experimental evaluation paragraph cannot be publish without any major modifications.
(1-1)
First of all the experiment is not clearly detailled:
- what is the temperature change?
(Reply)
The temperature are measured from 4 am to 1pm every 1 minute in the last ten days of April. The measurement day was sunny. Since an automatic air conditioner is stopped, the cause of changing temperature is a climate shift, such as the sunshine condition, the wind condition, and so on.
In the revised paper, the detail of measurement is shown.
(1-2)
- figure 13 is a simulation result not an experimental one. Because a part of the theoretical results have been yet published, the experimental part should be the added value of the paper. So the paper cannot be publish as soon as this experiment is fully realized and well explained. Experimental transmission must be realized.
Actually the paper don't provide any added value compare to the paper published in IEEE Wireless Communications and Networking Conference in 2017 and entitled :
[A1] Data Tracking Using Frequency Offset and SIC for Physical Wireless Conversion Sensor Networks 7 Author(s) Takehiro Sakai ; Osamu Takyu ; Keiichiro Shirai ; Mai Ohta ; Takeo Fujii ; Fumihito Sasamori ; Shiro Handa
The paper needs major revisions. Complete experiments should be made to validate the theoretical results.
(Reply)
The purpose of experimental measurement is clarifying the accuracy of sensing result separation by proposed scheme in performing to the actual sensing results. A data tracking of proposed scheme uses the three kinds of criterion, sensing result, channel state information (CSI), and carrier frequency offset (CFO). The time continuity of them decides the accuracy of data tracking. The time correlation of CSI is modeled in [A2]. In addition, the CFO is added to the transmit signal by sensor but the CFO is unintentionally shifted by the unstable of local oscillator [A3]. In this paper, the model of time correlation for CFO is used by the eq. (21) and eq.(22) of the revised paper.
[A2] J. G. Proakis, Digital communications, McGraw-Hill, fourth edition, 2001
[A3] B. Razavi, RF Microelectronics, Prentice Hall, 2012.
However, the time correlation of sensing results is different for the kinds of sensor. The performance evaluation with using the practical sensing results is important for clarifying the practicality of proposed scheme. The temperature sensor has not only time correlation but also the correlation among the sensors in different location. The correlation among sensors is related to the duplication of sensing results among sensors. This paper proposes the data tracking with using the criterion of frequency spectrum for the countermeasure of the time duplication. Therefore, the temperature sensor is reasonable for clarifying the recovering to the duplication of sensing results in practical environment.
In revised paper, one more evaluation result with the different temperature sensing result is added. The data of it is given in [A3]. In this evaluation, the heater is used as the heat source and the temperature is increased by short time. Since the difference of temperature among sensors is larger, the duplications of sensing results is fewer and does not occur, continuously. In the evaluation of data separation, the proposed separation with the lower time continuity of CSI and CFO achieves the more highly accuracy of data separation. Although the time correlation of CSI and CFO becomes low, the better data tracking is achieved owing to the highly time continuity of sensing results. From the evaluations with using actual sensing results, our proposed technique achieves the better data separation with the lower correlation of sensing results among sensors and the more highly time correlation of sensing results.
[A3]  R.Myoenzono, O.Takyu, K.Shirai, T.Fujii, M.Ohta, F.Sasamori, and S.Handa, ``Data tracking and effect of frequency offset to simultaneous collecting method for wireless sensor networks,'' Int. J. Distri. Sensor Netw., vol.2015, pp.1--10, Aug. 2015.
In the revised paper, the section title is revised from ``Experimental Evaluation’’ to `` Performance Evaluation based on Actual Sensing Results’’. In addition, the additional experimental evaluation is added. The revised paper also shows the full experimental evaluation whose data transmission is constructed by the experiment is important future works.
The progress from conference paper [A1] is the data tracking based on the frequency spectrum given by sections 4.3, 4.3.1., and 4.3.2. Ref. [A1] assumes the data duplication does not occur or occurs by only one time. Although the sensing data duplication causes the degradation of estimating CSI and CFO, Ref[A1] does not consider the countermeasure to it. In the numerical results of our paper, ``w/o adaptation’’ is the same as the scheme of Ref. [A1]. The proposed scheme, ``w/ adaptation’’, which includes the data tracking with frequency spectrum, achieves the better performance than ``w/o adaptation’’.
In revised paper, we add the footnote indicating that ``w/o adaptation’’ is the evaluation by the scheme of Ref. [A1] for clarifying the progress of our paper.
Thank you very much for your important suggestion.

Reviewer 2 Report
The paper introduces an ID insertion technique in the signal received by a fusion center from a sensor. The applied technique deals with a method of sensing result separation focused on fractional carrier frequency offset (CFO) for PhyC-SN. The paper is well-written and describes the applied methodology to assign an ID to the received signal. The technique validity is shown through computer simulation by evaluating the accuracy of the proposed data separation and good results were reported. However, the authors do not show a comparison of results with respect to other data separation techniques by using the same simulation environment of this paper.
Author Response
Reply Sheet
Dear Dr. Reviewer 2
Thank you very much for spending your precious time to review our paper.
Thanks to your important comments, the revised paper is much better than the original one. Please, re-review our paper, again. We would like to show the answer to your comments.
(2-1)
The paper introduces an ID insertion technique in the signal received by a fusion center from a sensor. The applied technique deals with a method of sensing result separation focused on fractional carrier frequency offset (CFO) for PhyC-SN. The paper is well-written and describes the applied methodology to assign an ID to the received signal. The technique validity is shown through computer simulation by evaluating the accuracy of the proposed data separation and good results were reported. However, the authors do not show a comparison of results with respect to other data separation techniques by using the same simulation environment of this paper.
(Reply)
A time hopping scheme has been proposed for conventional data separation [B1]. It consumes the time slots for sending ID but our proposed scheme does not. The fair comparison between them is difficult. The data tracking scheme based on Kalman filtering has also been proposed [B2]. The numerical result of it is shown in ``Energy’’ of the numerical results of the revised paper.
[B1] S. Sakai and T. Fujii, ``Performance of physical wireless parameter conversion sensor network ({PHY-C SN}) using frequency hopping,'' Proc. IEEE Int. Conf. Netw. Infra. Digital Content (IC-NIDC), pp.293--296, Sept.\ 2014.
[B2] R.Myoenzono, O.Takyu, K.Shirai, T.Fujii, M.Ohta, F.Sasamori, and S.Handa, ``Data tracking and effect of frequency offset to simultaneous collecting method for wireless sensor networks,'' Int. J. Distri. Sensor Netw., vol.2015, pp.1--10, Aug. 2015.
From the results, the conventional scheme degrades the accuracy of data separation because the data duplication causes the error tracking, which the fusion center wrongly recognizes the information source of sensing result as the different sensor.
In the revised paper, we add ``Energy’’ is the result of Ref [B2] to the footnote of the revised paper.
Thank you very much for your important suggestion.

Reviewer 3 Report
The authors provide good method to insert ID for sensors in collecting data. All the results seem fine to the reviewer. The authors review well existing papers in the field.
The manuscript could improved better if the authors compare their method to the other papers in the literature review.
Author Response
Reply Sheet
Dear Dr. Reviewer 3
Thank you very much for spending your precious time to review our paper.
Thanks to your important comments, the revised paper is much better than the original one. Please, re-review our paper, again. We would like to show the answer to your comments.
(3-1)
The authors provide good method to insert ID for sensors in collecting data. All the results seem fine to the reviewer. The authors review well existing papers in the field.
The manuscript could improved better if the authors compare their method to the other papers in the literature review.
(Reply)
A time hopping scheme has been proposed for conventional data separation [C1]. It consumes the time slots for sending ID but our proposed scheme does not. The fair comparison between them is difficult. The data tracking scheme based on Kalman filtering has also been proposed [C2]. The numerical result of it is shown in ``Energy’’ of the numerical results of the revised paper.
[C1] S. Sakai and T. Fujii, ``Performance of physical wireless parameter conversion sensor network ({PHY-C SN}) using frequency hopping,'' Proc. IEEE Int. Conf. Netw. Infra. Digital Content (IC-NIDC), pp.293--296, Sept.\ 2014.
[C2] R.Myoenzono, O.Takyu, K.Shirai, T.Fujii, M.Ohta, F.Sasamori, and S.Handa, ``Data tracking and effect of frequency offset to simultaneous collecting method for wireless sensor networks,'' Int. J. Distri. Sensor Netw., vol.2015, pp.1--10, Aug. 2015.
From the results, the conventional scheme degrades the accuracy of data separation because the data duplication causes the error tracking, which the fusion center wrongly recognizes the information source of sensing result as the difference sensor.
In the revised paper, we add ``Energy’’ is the result of Ref [C2] to the footnote of the revised paper.
Thank you very much for your important suggestion.

Reviewer 4 Report
Authors propose the insertion technique of ID with carrier frequency offset to each transmit signal and the sensing data separation with the inter carrier interference cancellation and the data tracking technique for a physical wireless parameter conversion sensor networks . The fractional with carrier frequency offset is assigned to each user for specifying the sensor node.
They also proposes the construction of inter carrier interference replica in a narrow band wireless communication system considering the estimation of the channel state information and the with carrier frequency offset from the received signals
The inter carrier interference cancellation of the narrowband wireless communications is proposed. The two types of data tracking techniques are proposed and are used by the fusion center.
Some suggestions
The paper is well structured presenting a good introduction to the topic, as well as an adequate approach and description of the background. However, next I indicate some comments and suggestions
1.Perhaps references dating from 2006, 2009, 2011 are no longer so relevant and could be removed.
2.Overview of PhyC-S is dealt with in detail, but references could be used to avoid including well known theoretical aspects.
3. Proposed Sensing Results Separation is well thought out and developed. Although perhaps item 4.2 could be included in the earlier or later one, as a commentary, and remove it.
4.Numerical results and experimental result are presented in the work. It is missed that the tests are carried out in different configurations of scenario and test conditions; fundamentally in the empirical tests, where more test must be included to be able to validate the hypotheses.
5.The work profusely describes the benefits of the proposal, but does not indicate those aspects that represent weak points or restrictions, which are of as much interest as those for all readers who investigate in the same topic.
Author Response
Reply Sheet
Dear Dr. Reviewer 4
Thank you very much for spending your precious time to review our paper.
Thanks to your important comments, the revised paper is much better than the original one. Please, re-review our paper, again. We would like to show the answer to your comments.
Authors propose the insertion technique of ID with carrier frequency offset to each transmit signal and the sensing data separation with the inter carrier interference cancellation and the data tracking technique for a physical wireless parameter conversion sensor networks . The fractional with carrier frequency offset is assigned to each user for specifying the sensor node. They also proposes the construction of inter carrier interference replica in a narrow band wireless communication system considering the estimation of the channel state information and the with carrier frequency offset from the received signals. The inter carrier interference cancellation of the narrowband wireless communications is proposed. The two types of data tracking techniques are
proposed and are used by the fusion center.
Some suggestions
The paper is well structured presenting a good introduction to the topic, as well as an adequate approach and description of the background. However, next I indicate some comments and suggestions
(4-1) Perhaps references dating from 2006, 2009, 2011 are no longer so relevant and could be removed.
(Reply)
The revised paper remove the suggested three references and the related sentences.
Thank you very much for your important suggestion.
(4-2) Overview of PhyC-S is dealt with in detail, but references could be used to avoid including well known theoretical aspects.
(Reply)
The section 4, which explains the proposed scheme, refers the equation derived by section 3. Since the parameters of reference papers are different from our paper and the derivation of received signal is required for showing how to construct the inter-carrier interference replica, the revised paper also shows the theoretical analyzes of section3. In revised paper, the explanation of the impact of carrier frequency offset is removed because Ref. [D1] explains it. Instead, the revised paper refers Ref. [D1].
[D1] R.Myoenzono, O.Takyu, K.Shirai, T.Fujii, M.Ohta, F.Sasamori, and S.Handa, ``Data tracking and effect of frequency offset to simultaneous collecting method for wireless sensor networks,'' Int. J. Distri. Sensor Netw., vol.2015, pp.1--10, Aug. 2015.
Thank you very much for your important suggestion.
(4-3) Proposed Sensing Results Separation is well thought out and developed. Although perhaps item 4.2 could be included in the earlier or later one, as a commentary, and remove it.
(Reply)
The proposed separation adaptively changes the two protocol in the duplication of sensing results or not. In original paper, section 4.2 is independent to the other section but in revised paper, ``Sensing results separation without duplication of sensing results’’ includes three subsections ``spectrum detection and estimating CSI and CFO’’ and ``Construction of ICI replica and process of interference cancellation’’, and ``Data Tracking for Sensing Result Separation’’.
In addition, the section ``Sensing results separation without duplication of sensing results’’ is wrong but `` Sensing results separation with duplication of sensing results’’ is true. It is revised.
Thank you very much for your important suggestion.
(4-4) Numerical results and experimental result are presented in the work. It is missed that the tests are carried out in different configurations of scenario and test conditions; fundamentally in the empirical tests, where more test must be included to be able to validate the hypotheses.
(Reply)
In the revised paper, the detail of experimental environment is explained. In addition, the performance with new sensing results is added to the revised paper. The sensing results are evaluated by Ref. [D1]. The difference between the sensing results of original paper and that of Ref. [D1] is heater. The former does not uses the heater and obtains the temperature by every 1 minute and from 4am to 1pm in the last ten days of April. The latter uses the heater and obtain the temperature by every 1 seconds during 500 secs in the end of November.
In the newly added sensing results, since the difference of temperature among sensors is larger, the duplications of sensing results are fewer and do not occur, continuously. In the evaluation of data separation, the proposed separation with the lower time continuity of CSI and CFO achieves the highly accuracy of data separation. Although the time correlation of CSI and CFO becomes low, the better data tracking is achieved owing to the highly time continuity of sensing results. From the evaluations with using actual sensing results, our proposed technique achieves the better data separation with the lower correlation of sensing results among sensors and the more highly time correlation of sensing results.
In the revised paper, the evaluation system composed of the experiment and the computer simulation is explained in detail. In addition, the additional experimental evaluation is added.
Thank you very much for your important suggestion.
(4-5) The work profusely describes the benefits of the proposal, but does not indicate those aspects that represent weak points or restrictions, which are of as much interest as those for all readers who investigate in the same topic.
(Reply)
The proposed separation and ID insertion have two disadvantages. If the data duplication continuously and frequently occur, the accuracy of separation is degraded because it continuously uses the past estimated CSI and CFO and thus the difference between the past estimated CSI and CFO and the practical ones becomes larger. It is one of disadvantages. In second disadvantages, the process of adding the fractional frequency offset to carrier signal is complicated for the sensor. Recovering these disadvantages is important future works.
In revised paper, we shows the two disadvantages in conclusion.
Thank you very much for your important suggestion.

Round 2
Reviewer 1 Report
Despite the wireless communication process is realized by computer simulation, the conditions and parameters of the experimental measurement results are clearly defined.
The authors have clearly separated the simulation from the experimental results and provided a much more comprehensible paper.
A good discussion of the results provided in figure 13 and 14 has been done together with a clear presentation of the two disadvantages of the proposed technique inside the conclusion.
This paper provide a good added value compare to the later published papers.
The authors have finished off the study, taken care of the previous remarks and provided the required information necessary for a good understanding of the article.
